# Research Trends in the Study of Acceptability of Digital Mental Health-Related Interventions: A Bibliometric and Network Visualisation Analysis

Maria Armaou 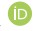

School of Health Sciences, University of Nottingham, Nottingham NG7 2HA, UK;
maria.armaou@nottingham.ac.uk

**Abstract:** The acceptability of digital health interventions is a multifaceted concept that is central to user engagement. It is influenced by cultural and social norms and it is, also, a key consideration for intervention development and evaluation. For this reason, it is important to have a clear overview of how research in digital interventions' acceptability has evolved, what type of measures or assessments have been most frequently utilised, and what may be the implications for the knowledge area and future research directions. The purpose of this bibliometric and network visualization analysis was to explore the main research patterns in the study of the acceptability of digital mental health interventions and highlight the key characteristics of knowledge production on this topic. The Web of Science was searched for relevant primary studies, with 990 documents selected for inclusion in this bibliometric analysis. Publications' metrics, text and author keyword analysis, and bibliographical coupling of the documents provided insights into how technological developments, specific research interests, research priorities, and contexts have shaped research in the field. The main differentiation in acceptability approaches emanated from the studies' research designs, the stage of intervention development and evaluation, and the extent to which there was a focus on user attitudes, experience, and engagement. These differentiations further indicate the importance of having clarity as to what concepts or elements of acceptability a study addresses as well as approaches that have the potential to address the complexities of acceptability.

**Keywords:** intervention acceptability; digital mental health interventions; bibliometric analysis

## 1. Introduction

The examination of intervention acceptability is considered necessary in the iterative process of the design, implementation, and evaluation of complex interventions (Skivington et al. 2021). Digital mental health interventions (DMHIs) are complex and they can include a variety of therapeutic components, be delivered via different technologies, be self-guided or offer different types of support, offer access to treatment, and aim to prevent poor mental well-being in the general population (Gega et al. 2022; Renfrew et al. 2021). There is an array of definitions of acceptability including "social acceptability", "treatment acceptability", and acceptability frameworks, while other studies have explained their approach to acceptability and how it is measured within their research design (Park et al. 2022; Sekhon et al. 2017). As a result, acceptability is frequently addressed in interventions at different stages of development and evaluation with authors adopting different approaches and assessment methods.

Multiple reviews of studies reporting digital mental health interventions have shown that, although there is evidence for their effectiveness, their main shortcoming is poor patient engagement (Lipschitz et al. 2023). Such findings have driven calls to develop clear criteria in reporting on intervention engagement (Lipschitz et al. 2022, 2023). At the same time, low engagement levels with digital mental health interventions and high attrition lead to a loss of participants in clinical trials, which reduces the prospect of 'digital therapeutics'

to complete regulatory trials (Nwosu et al. 2022). For this reason, there is a growing need for pragmatic studies to evaluate the usefulness of digital mental health tools (Hekler et al. 2016; Torous and Haim 2018).

Digital mental health is frequently hailed as an opportunity to transform mental health services, providing instantaneous and at-scale access to tailored mental health support across diverse populations and contexts (Hunter et al. 2023; Roland et al. 2020). Many healthcare systems across the world have been turning to telehealth as a means to boost access to mental health services, a trend ever increased following the response to the COVID-19 pandemic (Adepoju 2020; The Scottish Government 2021). However, emerging evidence also highlights the importance of addressing persistent gaps in access to mental health treatment across different demographics within high-, low-, and middle-income countries (Lu et al. 2022). Specific challenges can involve barriers faced by vulnerable groups in accessing digital health services, which can include communication difficulties and inadequate formal and informal support for service users (Gama et al. 2022; Goodman et al. 2021; Kaihlanen et al. 2022). At the same time, sociocultural diversity, poor infrastructure, and low bandwidth can limit intervention adoption (Banerjee et al. 2021).

Intervention acceptability is a separate concept from intervention engagement, but their close relationship forms a dynamic relationship that pertains to users' attitudes, users' engagement with the intervention, and intervention usability (Perski and Short 2021). For example, patients' engagement with digital mental health interventions (DMHIs) has been defined in terms of their adoption rates and the degree of users' sustained interactions with them (Arnold et al. 2021; Borghouts et al. 2021). Studies on the adoption of digital mental health interventions often deploy technology acceptance theories, such as the Technology Acceptance Model, precisely because they address the impact of technology acceptance constructs (i.e., perceived usefulness and perceived ease of use) on users' intentions to use technology (Sawrikar and Mote 2022). As shown by previous research, an assessment of users' intentions to use technology can be a predictor of the adoption of mHealth services and e-mental health for different stakeholders (Alam et al. 2021; De Veirman et al. 2022; Semwanga et al. 2021). Perki and Short's dynamic model of intervention acceptability demonstrates the complex relationship between acceptability and user engagement (Perski and Short 2021). The key point in Perki and Short's model is that acceptability is defined as an emergent outcome of the relationship between people's cognitions, beliefs, and affective responses; these include individuals' evaluation of the intervention's potential burden, usefulness, and fitness within their own value systems. Within this model, an individual's sociocultural contexts are considered to have a significant influence on people's beliefs that in turn shape and are shaped by user needs. This approach is reflected in previous reviews of the literature on barriers and facilitators of user engagement with digital health interventions (Borghouts et al. 2021; Gauthier-Beaupré and Grosjean 2023). A systematic review of user engagement in DMHIs for common mental health problems identified three clusters of user-centered, program-related, and technology- and environment-related factors that can influence user engagement (Borghouts et al. 2021). User-centered factors included demographic variables, personal traits, beliefs, mental health status, and previous experiences with technology. Program-related constructs included type of content, perceived fit with the user's culture, level of guidance, social connectedness, and impact of the intervention. Finally, technology- and environment-related constructs included cost and usability, privacy and confidentiality, social influence, and implementation factors (Borghouts et al. 2021). Similarly, a meta-ethnographic study on the acceptability of digital health technologies among francophone-speaking communities across the world showed how the perspectives and attitudes of different stakeholders towards different applications of technology (mobile technologies, robot technologies, telemedicine, sensors, and wearable technologies) can determine their willingness and commitment to use those technologies (Gauthier-Beaupré and Grosjean).

With the global digital mental health market size valued at USD 19.5 billion in 2022 and with a projection of it reaching USD 72.3 billion in 2032 (Gotadki 2024), it is vital to

understand the trends of knowledge production that pertain to intervention acceptability. A bibliometric analysis allows an overview of knowledge patterns and identifies knowledge gaps and emergent research fields, which is invaluable, especially when there is a fragmented area of knowledge (Hernández-Torrano et al. 2020; Skute et al. 2019).

## 2. Objectives

The aim of this study was to explore the knowledge production patterns associated with the concept of acceptability of digital mental health-related interventions in primary research studies. For this reason, the study objectives were to:

(a) Analyse the evolution of publications of primary studies from 2008 to 2023.
(b) Analyse the main contributors to the knowledge area.
(c) Report on authors' approaches to intervention acceptability based on an analysis of the documents' bibliometric features.

## 3. Methods

### 3.1. Data Sources and Search Strategy

All editions of citation indexes in the Web of Science Core Collection (WOSCC) were used to search articles in January 2024. The WOSCC was selected for this analysis as it is regarded as one of the most authoritative databases and because it is more selective than other databases and has greater keyword sensitivity (Niel et al. 2015). Although its overall coverage is smaller than that of Scopus, the use of only one database allows for a better quality of citation analysis (Caputo and Kargina 2022; Gusenbauer 2022).

The search period was set from January 2008 to December 2023. The year 2008 was selected as that was the first year following the release of the first iPhone. Searches included all primary research studies in all languages. The following query was used to search for relevant topics:

TS = (digital or online or on-line or internet-based or internet* or web-based or web* or computer-based or app* or smartphone or mobile* or mobile-assisted or computer* or wearable or virtual or chatbot or augmented reality) AND

DTS = (mental health or mental disorder or psychological wellbeing or mental wellbeing or mental well-being or mental illness)

AND TS = (acceptability or usability)) NOT (ALL = (protocol) OR ALL = (meta-analysis) OR ALL = (REVIEW) OR ALL = (health records))

TS in the WOS collection includes searches in titles, abstracts, and keywords.

The papers included were articles or proceeding papers. On the other hand, papers were excluded if they were any of the following: book chapters, review or editorial papers, retracted publications, book reviews or corrections, item withdrawals or retractions, withdrawn publications, or data papers.

The keyword searches for papers addressing the concept of acceptability included only the keywords "acceptability" or "usability" as they represent two distinct but greatly interlinked generic concepts. This strategy allowed the inclusion of empirical studies that discuss different ways in which intervention acceptability has been operationalised.

### 3.2. Eligibility Criteria

Titles and abstracts were screened by applying the following inclusion criteria: study population (any), study design (any), intervention (any digital intervention), outcomes (any). Documents were excluded (a) if they did not make explicit reference to digital tools or interventions, (b) if they reported a digital intervention that was not mental health related either in the relation to the population or its outcomes, (c) if they reported on analysis of health records, or (d) if they only reported on product capabilities or characteristics. Furthermore, review papers and theoretical pieces were excluded.

*3.3. Data Selection*

All selected papers were saved as "a marked list" on the researcher's WOS account. The final screening included checking the content of this list to exclude any duplicates and irrelevant papers. A final list of the selected documents was saved as a .txt document.

*3.4. Analysis*

The analysis was conducted via WOS statistics features, while VOSviewer (V. 1.6.19) was used to elicit tables and visualisation of text and keywords' co-occurrence, citations and publications, and documents' bibliographic coupling. Bibliographic coupling is a similarity measure that assesses the frequency in which specific documents are cited together in other publications and, as a result, it is an indicator of shared research themes.

An early review of the selected documents showed that the terms "acceptability" and "usability" were always present in the selected documents' title but frequently, the abstract and the keywords did not provide adequate information on how those two concepts were assessed. For this reason, the content of 10% (*n* = 100) of all selected documents was checked, and specific information was extracted on how those concepts were specifically addressed in each document which informed the analytic strategy for a comprehensive review of the documents' bibliometric features.

## 4. Results

(A)   Descriptives

A total of 2514 references were identified in the WoS from 2008 to 2023 and were screened based on the study's inclusion and exclusion criteria. In total, 990 documents were selected from 5598 authors, 1672 institutions, and 78 countries. Figure 1 shows the publication and citation rates. The year 2022 has the peak publication number (192) and citation number (3020), followed by 2023, with 191 documents having 3131 total citations.

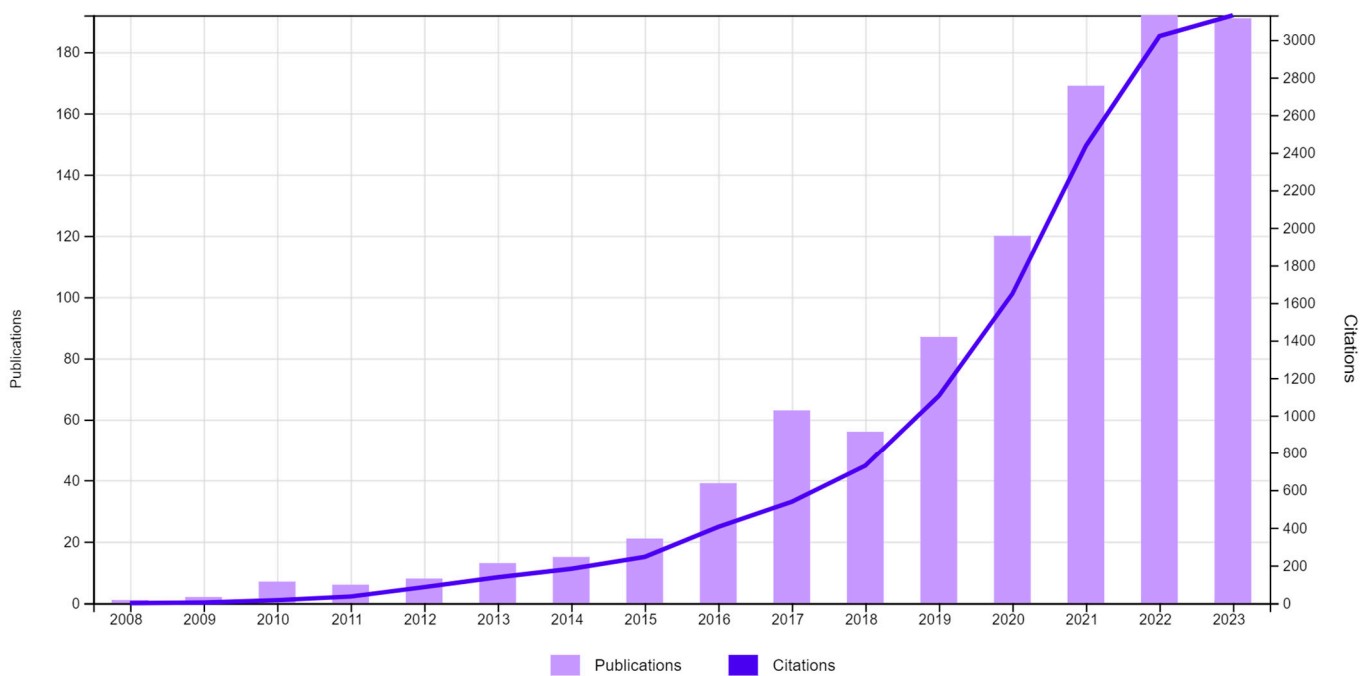

**Figure 1.** Times cited and publications over time.

(B)   Analysis of knowledge production

A co-authorship analysis was conducted to identify the most productive research authors, research teams, journals, and organisations.

### 4.1. Authors

Of the 5598 authors, 36 produced at least five papers. Table 1 shows the top 10 most productive authors of the extracted documents. The most productive author was Nick Titov with 14 papers from Macquarie University in Australia, followed Blake F. Dear from Macquarie University, David C. Mohr from Northwestern University in USA, Mario Alvarez-Jimenez from the University of Melbourne, and Gerhard Andersson from Linköping University in Sweden, with 11 papers each.

**Table 1.** Top 10 authors.

|    | Author | Documents | Countries |
|----|--------|-----------|-----------|
| 1  | Nickolai Titov | 14 | Australia |
| 2  | Blake F. Dear | 11 | Australia |
| 3  | David C. Mohr | 11 | USA |
| 4  | Mario Alvarez-Jimenez | 11 | Australia |
| 5  | Gerhard Andersson | 11 | Sweden |
| 6  | Helen Christensen | 10 | Australia |
| 7  | Pim Cuijpers | 9 | Vrije Universiteit Amsterdam |
| 8  | Dror Ben-zeev | 8 | USA |
| 9  | Helen Riper | 8 | Vrije Universiteit Amsterdam |
| 10 | Gavin Andrews | 7 | Australia |

Figure 2 depicts collaboration maps among the key authors who had at least five documents. The colours represent working groups, the size of the circle depicts the number of articles published by each author, and the lines represent the total strength of the co-authorship between authors. It shows 15 clusters of authors from different countries. Three clusters of researchers (yellow, green, and red clusters) include authors from Australian institutions, while the purple cluster includes authors from USA-based institutions.

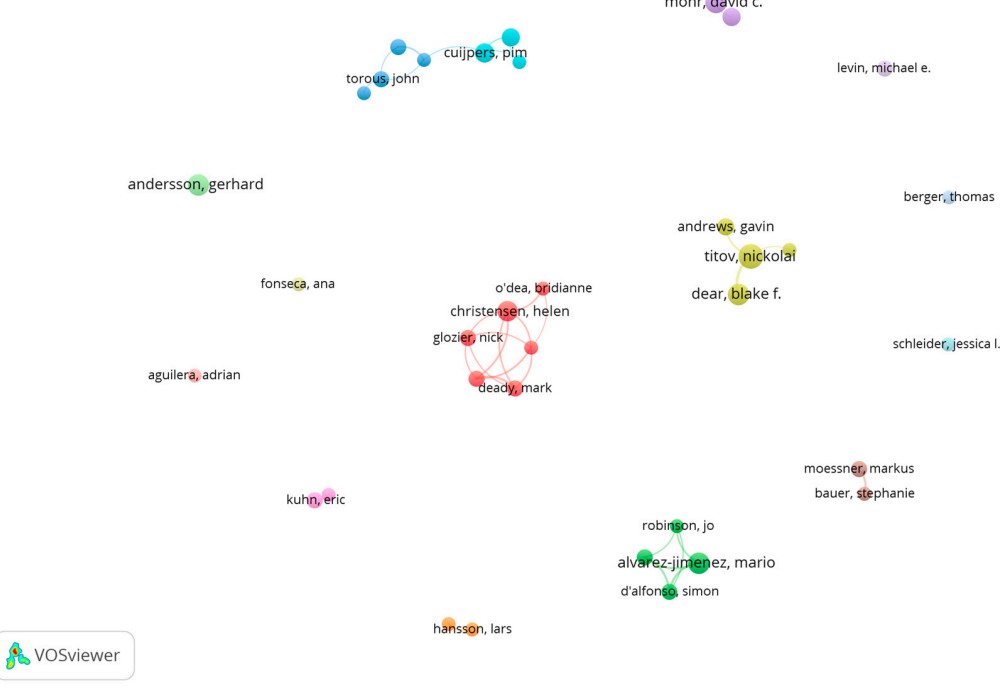

**Figure 2.** Authors' collaboration maps.

Among those, there was only one cluster of consistent co-authorships between European and USA-based working groups. It consisted of the working group in red, which included Pim Cuijpers and Heleen Riper from the Vrije University of Amsterdam and David and Daniel Ebert from the Technische Universität München (TUM). Pim Cuijpers also collaborated with Dr. Patel Vlkram from Harvard University, while John Torous from Harvard University collaborated with Sandra Bucci from the University of Manchester.

*4.2. Journals*

The most popular sources were JMIR Formative Research with 89 records, JMIR mental health with 72 records, and Journal of Medical Internet Research with 59 records. JMIR Mental Health was ranked first in citations, with 1893, followed by the Journal of Medical Internet Research with 1472 citations, and PlOS One with 784 citations (Table 2).

**Table 2.** Top ranking knowledge contributors.

| Ranking | Publication Titles | Record Count | Citations | Affiliations | Record Count | Citations | Countries | Record Count | Citations |
|---|---|---|---|---|---|---|---|---|---|
| 1 | JMIR Formative Research | 89 | 210 | University of Melbourne | 46 | 800 | USA | 391 | 5737 |
| 2 | JMIR Mental Health | 72 | 1893 | Kings College London | 41 | 638 | Australia | 193 | 3671 |
| 3 | Journal of Medical Internet Research | 59 | 1472 | University of Sydney | 34 | 492 | England | 163 | 1955 |
| 4 | Internet Interventions-The Application of Information Technology in Mental and Behavioural Health | 39 | 378 | University of Washington | 31 | 506 | Germany | 71 | 805 |
| 5 | International Journal of Environmental Research and Public Health | 28 | 240 | University of New South Wales | 29 | 950 | Canada | 70 | 475 |
| 6 | JMIR mHealth and uHealth | 24 | 617 | Stanford University | 24 | 252 | Netherlands | 67 | 880 |
| 7 | Frontiers in Psychiatry | 22 | 266 | Northwestern University | 22 | 381 | Sweden | 39 | 461 |
| 8 | Mindfulness | 14 | 116 | Monash University | 22 | 327 | Spain | 37 | 400 |
| 9 | Journal of Affective Disorders | 12 | 327 | Vrije University of Amsterdam | 21 | 244 | People's Republic of China | 26 | 206 |
| 10 | BMC Psychiatry | 12 | 317 | University of Oxford | 20 | 161 | New Zealand | 22 | 289 |

### 4.3. Institutions

The most productive institutions were the University of Melbourne with 46 records, Kings College London with 41 records, and University of Sydney with 34 records. The University of New South Wales was ranked first in citations, with 950, followed by the University of Melbourne with 800 citations, and King's College London with 638 citations (Table 2).

### 4.4. Countries

The most productive countries were the USA with 391 records and 5737 citations, followed by Australia with 193 records and 3671 citations, and England with 163 records and 1955 citations (Table 2).

(C)  Analysis of authors' approaches to the acceptability of digital mental health-related interventions

### 4.5. Text Co-Occurrence Analysis

A co-occurrence analysis of text found at least ten times across all documents' titles and abstracts was conducted in order to obtain a broad overview of key areas of interest and further inform the data analysis and synthesis strategy. A binary frequency was applied, which means that the presence or the absence of a term in a document was counted, but the number of times the term was used in each document was not counted. Of the 22,942 terms, 734 met the set threshold and, subsequently, the most relevant terms were selected to calculate a relevance score. The default choice of VOSviewer is to select the 60% most relevant terms, which is what was selected in this study, with 440 terms included in the analysis.

Figure 3 shows three broad areas of interest. The larger cluster of items in red ($n = 202$) corresponds to intervention development and implementation. Within this cluster, the most frequent terms were "development" ($n = 244$), "user" ($n = 176$), "technology" ($n = 156$), and "implementation" ($n = 142$). Other terms in the same cluster were "perspective" (77), "feature" ($n = 99$), "perception" ($n = 77$), "focus group ($n = 69$), "knowledge" ($n = 93$), "application" ($n = 93$), and "recommendation" ($n = 67$).

The second cluster in green with 169 items corresponds to intervention outcomes. In the green cluster, the most frequent terms were "depression" ($n = 322$), "anxiety" ($n = 249$), "week" ($n = 249$), and "baseline" ($n = 195$), and "internet" ($n = 175$). Other terms within the same cluster were "effect" ($n = 184$), "trial" ($n = 193$), "effect" ($n = 184$), adherence ($n = 107$), and "significant improvement" ($n = 75$).

Finally, the third smaller cluster in blue with 71 items corresponds more to intervention and sample characteristics. In that cluster, the most frequent terms within the smaller blue cluster were "female" ($n = 55$), "respondent" ($n = 47$), and "characteristic" ($n = 45$). Other terms within the same cluster were "mental health support" ($n = 36$), "primary care" ($n = 31$), "male", and "suicide" ($n = 27$).

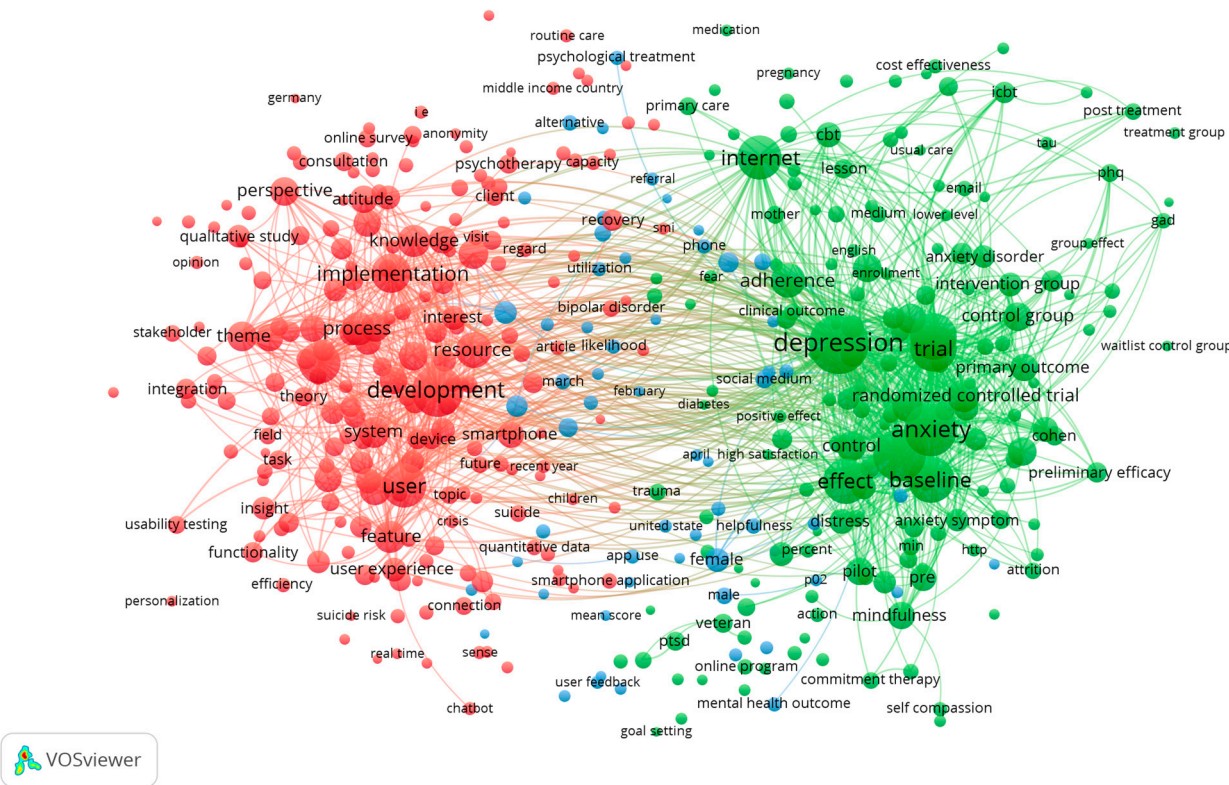

**Figure 3.** Clusters of Text Co-occurrence in documents' titles and abstracts.

*4.6. Authors' Keywords Co-Occurrence Analysis*

Overall, three analyses of authors' keywords were conducted. In the first one, all authors' keywords were included to obtain a broad view of authors' key research interests. The most frequent keywords were "mental health" ($n$ = 264), "depression" ($n$ = 168), "mhealth" ($n$ = 107), "anxiety" ($n$ = 93), "mobile phone" ($n$ = 76), and "acceptability" ($n$ = 56). However, the network analysis (Figure 4) did not provide comprehensive information about the co-occurrence patterns of keywords connected with intervention acceptability. For this reason, two separate analyses of authors' keywords were conducted. The analyses included manually identified and selected keywords within the Vosviewer system, with an occurrence frequency of at least two. Of 2287 keywords, 643 met that threshold. The first analysis reported on keywords that occurred at least twice and were associated with digital technologies and their application to the delivery of mental health interventions. Subsequently, those were mapped according to the documents' publication years. The second analysis reported on keywords that were relevant to acceptability or usability, research design, and research context or population characteristics.

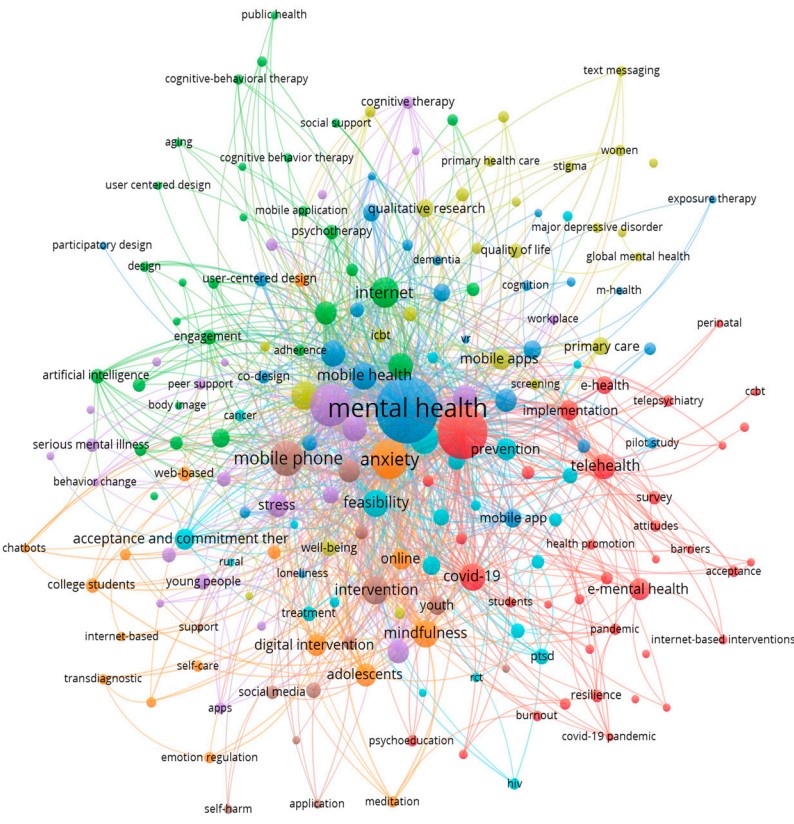

**Figure 4.** Network analysis of all author keywords.

*4.7. Digital Technology Application in the Delivery of Mental Health Intervention Studies*

Of the 643 keywords, 115 relevant keywords were manually identified and included in the analysis. Table 3 shows relevant authors' keywords with a frequency of at least 10. The keyword "mhealth" was ranked first ($n = 107$), followed by "mobile phone" ($n = 76$) and "internet" ($n = 54$).

Figure 5 shows authors' keywords' occurrence frequencies mapped across the documents' average publication year. The earliest used keywords had low overall frequencies and included "smartphones" ($n = 4$, Avg. pub. Year: 2016.50), "internet-based treatment" ($n = 6$, Avg. pub. Year: 2017.33), and "m-health" ($n = 5$, Avg. pub. Year: 2017.80). The keyword "internet" ranked third overall ($n = 54$) and had an average publication year in early 2018 (Avg. pub. Year = 2018.17). Other keywords with an average publication year in 2018 included "telepsychiatry" ($n = 10$, Avg. pub. Year: 2018.89), "e-health " ($n = 19$, Avg. pub. Year: 2018.89), and "internet interventions"($n = 11$, Avg. pub. Year = 2018.82). Keywords relevant to the application of digital technology in mental health-related research with an average publication year in 2019 included "ehealth" ($n = 42$, Avg. pub. Year: 2019.74), "web-based" ($n = 14$, Avg. pub. Year = 2019.14), "internet-based" ($n = 6$, Avg. pub. Year: 2019.17), "smartphone" (Avg. pub. Year = 2019.78), "videoconferencing ($n = 7$, Avg. pub. Year: 2019.86), and "web-based intervention" (Avg. pub. Year: 2019. 88).

**Table 3.** Frequencies of technology-relevant keywords.

| Keywords | Occurrences | Keywords | Occurrences |
|---|---|---|---|
| mhealth | 107 | e-health | 19 |
| mobile phone | 76 | online intervention | 18 |
| internet | 54 | web-based intervention | 16 |
| digital health | 47 | mobile applications | 15 |
| mobile health | 45 | internet intervention | 14 |
| ehealth | 42 | internet-based intervention | 14 |
| telehealth | 38 | web-based | 14 |
| telemedicine | 37 | artificial intelligence | 13 |
| digital mental health | 33 | app | 12 |
| technology | 33 | digital | 12 |
| smartphone | 32 | conversational agent | 11 |
| digital intervention | 30 | icbt | 11 |
| e-mental health | 29 | internet interventions | 11 |
| online | 27 | chatbot | 10 |
| mobile apps | 26 | | |
| virtual reality | 22 | | |
| mobile app | 21 | | |

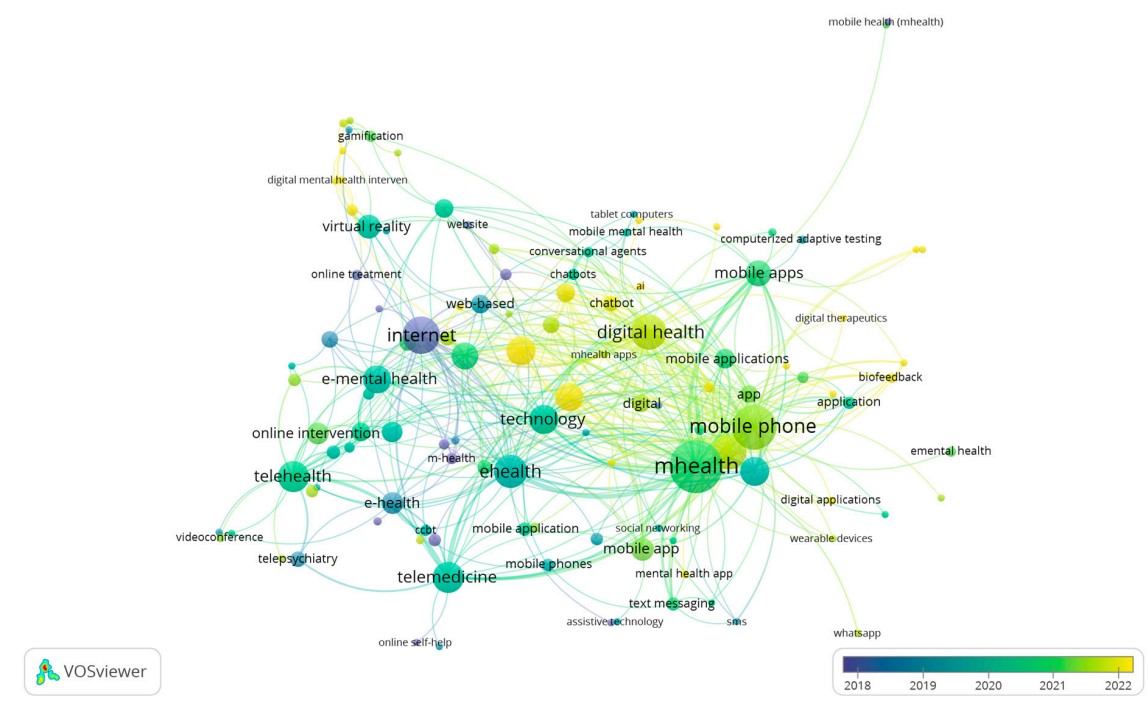

**Figure 5.** Technology keywords' publication year mapping.

Keywords' analysis shows that from 2020 onwards, interventions were more frequently described as mobile or digital interventions rather than online or internet-based interventions that coincided with advancements in technology and the need for digital delivery of interventions due to the COVID-19 pandemic. The most frequently occurring keywords with an average publication year in 2020 was "mhealth" (*n* = 107, Avg. pub. Year: 2020.82). Other keywords for the same year were "telehealth" (*n* = 38, Avg. pub.

Year: 2020.47), "technology" (*n* = 33, Avg. pub. Year = 2020.12), "e-mental health" (*n* = 29, Avg. pub. Year: 2020.00), "telemedicine" (*n* = 37, Avg. pub. Year = 2020.08), "mobile apps" (*n* = 26, Avg. pub. Year: 2020.73)","internet intervention" (*n* = 14, Avg. pub. Year: 2020.43), and " chatbots" (*n* = 5, Avg. pub. Year: 2020.40).

The most frequently occurring keyword with an average publication in 2021 was "mobile phone" (*n* = 76, Avg. pub. Year: 2021.34). Other keywords were "digital health" (*n* = 47, Avg. pub. Year = 2021.57), "mobile health" (*n* = 46, Avg. pub. Year: 2021.43), app" (*n* = 12, Avg. pub. Year = 2021.17), "online intervention" (*n* = 18, Avg. pub. Year: 2021.06), "conversational agent" (*n* = 11, Avg. pub. Year: 2021.55), "artificial intelligence" (*n* = 13, Avg. pub. Year: 2021.85), "chatbot" (*n* = 10, Avg. pub. YearL2021.80), "gamification" (*n* = 5, Avg. pub. Year: 2021.00), "digital technology" (*n* = 5, Avg. pub. Year_2021.00), "serious games" (*n* = 4, Avg. pub. Year: 2021.50). The most frequently occurring keywords in papers with an average publication in 2022 were "digital mental health" (*n* = 33, Avg. pub. Year = 2022.21) followed by "digital intervention" (*n* = 30, Avg. pub. Year = 2022.07), "vr" (*n* = 5; "Avg. pub. Year: 2022.80), and "biofeedback" (*n* = 4, Avg. pub. Year = 2022.00).

### 4.8. Acceptability and Usability Studies of Digital Mental Health-Related Interventions

Authors' approaches to intervention acceptability were examined by an analysis of the co-occurrence of keywords that (a) were related to assessments/approaches to intervention acceptability or usability, (b) referred to a study's research design, and (c) referred to specific target population characteristics. Keywords related to mental health outcomes were not selected in that analysis as their bibliometric features were assessed as part of the bibliographical coupling. Overall, 183 relevant keywords were identified and included in the analysis. Table 4 shows relevant authors' keywords with a frequency of at least five.

**Table 4.** Frequencies of authors' keywords relevant to intervention acceptability research.

| Keyword | Occurrences | Keyword | Occurrences | Keyword | Occurrences |
|---|---|---|---|---|---|
| Acceptability | 56 | Social media | 11 | Workplace | 7 |
| Feasibility | 45 | Survey | 11 | Clinical trial | 6 |
| COVID-19 | 45 | Young adult | 11 | Mixed methods | 6 |
| Usability | 42 | Feasibility study | 10 | Qualitative evaluation | 6 |
| Adolescent | 32 | University students | 10 | Qualitative study | 6 |
| Adolescents | 32 | Attitudes | 9 | RCT | 6 |
| implementation | 24 | Pandemic | 9 | Barriers | 5 |
| Veterans | 21 | Pilot study | 9 | Breast cancer | 5 |
| Youth | 21 | Pregnancy | 9 | Chronic illness | 5 |
| Qualitative research | 20 | Usability testing | 9 | COVID-19 pandemic | 5 |
| Suicide | 16 | Acceptance | 8 | Cultural adaptation | 5 |
| Young people | 15 | Adherence | 8 | Homelessness | 5 |
| College students | 14 | Children | 8 | Nurses | 5 |
| Engagement | 14 | Development | 8 | Participatory design | 5 |
| Qualitative | 14 | HIV | 8 | Perception | 5 |
| Caregivers | 13 | Women | 8 | Perinatal | 5 |
| User experience | 13 | Young adults | 8 | Postpartum period | 5 |
| User-centered design | 13 | Child | 7 | Rural | 5 |
| Co-design | 11 | Dementia | 7 | User centered design | 5 |
| Older adults | 11 | Design | 7 | Veteran | 5 |
| Parents | 11 | Students | 7 | | |

Figure 6 shows that author keywords were grouped in 10 clusters. The first cluster with 29 items had the keyword "COVID-19" (*n* = 45). This cluster did not include a keyword directly addressing acceptability or usability. It included relevant research designs such as "feasibility study" (*n* = 10) and "pilot study" (=9). Specific populations of interest were "health care workers" (*n* = 3), "healthcare provider" (*n* = 3), "nurses" (*n* = 3) and "African Americans" (*n* = 3).

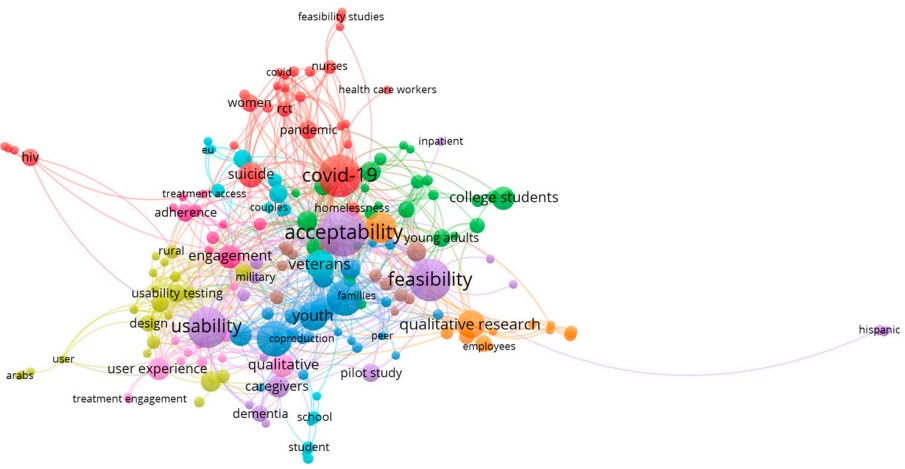

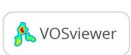

**Figure 6.** Acceptability research clusters.

The second cluster in green with 25 items had the keyword "veterans" (*n* = 21) and "college students" (*n* = 14). Keywords associated with acceptability or usability were "attitudes" (*n* = 9) and "barriers" (*n* = 5). Keywords related to research design approaches were "implementation science" (*n* = 4) and "clinical trials" (*n* = 6). Other populations of interest were "young adults" (*n* = 8), "children" (*n* = 8), "parent", "military" (*n* = 4), and "India" (*n* = 3).

The third cluster in blue with 23 items had the keyword "adolescents" (*n* = 32). This cluster did not have a specific keyword directly referring to acceptability and usability but had the most relevant research designs, including "user-centered design" (*n* = 13), "co-design" (*n* = 11), and "coproduction" (*n* = 3). Populations of interest included "youth" (*n* = 21), "cystic fibrosis" (*n* = 4), "teenager" (*n* = 3), "family caregivers" (*n* = 3), and "LGBTIQQ plus" (*n* = 3).

The fourth cluster in yellow with 22 items had the keyword "development" (*n* = 8) and "usability testing" (*n* = 9). Other keywords relevant to acceptability or usability were "usage" (*n* = 3), "needs assessment" (*n* = 2), and "user" (*n* = 2). Relevant research design keywords were "user-centered development" (*n* = 2), "participatory action research" (*n* = 2), "stakeholder participation" (*n* = 3), and "focus group" (*n* = 3). Relevant populations of interest were "older adults" (*n* = 11) and "rural" (*n* = 5).

The fifth cluster with 19 items in purple had the keywords "acceptability" (*n* = 57), "feasibility" (*n* = 45), "usability" (*n* = 42), and "pilot study" (*n* = 9). Other keywords in this cluster directly relevant to acceptability were "technology acceptance model" (*n* = 4) and "uptake" (*n* = 4). Populations of interest included "caregivers" (*n* = 13), "veteran" (*n* = 5), and "Hispanic" (*n* = 3).

The sixth cluster with 18 items in light blue had the keyword "veterans" (*n* = 22). The only keyword relevant to acceptability or usability within this cluster was "attitudes" (*n* = 9), and the only one relevant to research design was "qualitative methods" (*n* = 2).

Other populations of interest within this cluster were "pregnancy" (*n* = 9), "family" (*n* = 4) "breast cancer" (*n* = 5), and "school" (*n* = 3).

The seventh cluster with 12 items in orange had the keywords "qualitative research" (*n* = 20) and "implementation" (*n* = 24). Keywords that were relevant to acceptability were "acceptability of healthcare" (*n* = 4) and "attitude to computers" (*n* = 4), while keywords about populations of interest included "low- and middle- income countries" (*n* = 4), and "workplace" (*n* = 7).

The eighth cluster with 12 items in brown had the keywords "feasibility study" (*n* = 10) and "cultural adaptation" (*n* = 5). Another keyword relevant to research design was "participatory design" (*n* = 5), and populations of interest included "university students" (*n* = 10), "aboriginal" (*n* = 3), "indigenous" (*n* = 4), "first nations" (*n* = 3) "Indonesia" (*n* = 4), "refugees" (*n* = 4), and "South Africa" (*n* = 3).

The ninth cluster with 12 items in pink had the keyword "user experience" (*n* = 17). Some keywords relevant to intervention acceptability and its links with intervention engagement (*n* < 3) were "user feedback", "user satisfaction", app usability", "system usability", and "treatment engagement". Keywords relevant to research designs were "qualitative" (*n* = 14), "interview" (*n* = 2), "focus group" (*n* = 3), and "thematic analysis" (*n* = 4).

The tenth cluster with 10 items in fuchsia had the keyword "engagement" (*n* = 14). Keywords relevant to acceptability were "adherence" (*n* = 8), "uptake" (*n* = 4), "usability study" (*n* = 4), and "user engagement" (*n* = 4). It did not include specific keywords about the research design of the studies, while two populations of interest (*n* < 3) were "overweight" and "developing countries".

*4.9. Bibliographical Coupling*

A bibliographical coupling of documents was conducted to examine how often the same documents were present in the reference lists of other publications and thus identify key drivers of publication interests. Table 5 shows the 10 most cited documents. The most cited paper was Fitzpatrick et al.'s (2017) with 665 citations.

**Table 5.** Top 10 most cited publications.

| Ranking | Documents | Title | Citations |
|---|---|---|---|
| 1 | Fitzpatrick et al. (2017) | Delivering Cognitive Behavior Therapy to Young Adults With Symptoms of Depression and Anxiety Using a Fully Automated Conversational Agent (Woebot): A Randomized Controlled Trial | 655 |
| 2 | Donaghy et al. (2019) | Acceptability, benefits, and challenges of video consulting: a qualitative study in primary care | 242 |
| 3 | Titov et al. (2010) | Internet Treatment for Depression: A Randomized Controlled Trial Comparing Clinician vs. Technician Assistance | 235 |
| 4 | Ben-Zeev et al. (2013) | Development and usability testing of FOCUS: A smartphone system for self-management of schizophrenia. | 189 |
| 5 | Titov et al. (2013) | Improving Adherence and Clinical Outcomes in Self-Guided Internet Treatment for Anxiety and Depression: Randomised Controlled Trial | 173 |
| 6 | Huberty et al. (2019) | Efficacy of the Mindfulness Meditation Mobile App "Calm" to Reduce Stress Among College Students: Randomized Controlled Trial | 163 |
| 7 | Robinson et al. (2010) | Internet Treatment for Generalized Anxiety Disorder: A Randomized Controlled Trial Comparing Clinician vs. Technician Assistance | 162 |
| 8 | Musiat et al. (2014) | Understanding the acceptability of e-mental health—attitudes and expectations towards computerised self-help treatments for mental health problems | 150 |
| 9 | Comer et al. (2017) | Remotely delivering real-time parent training to the home: An initial randomized trial of Internet-delivered parent–child interaction therapy (I-PCIT). | 126 |
| 10 | Levin et al. (2017) | Web-Based Acceptance and Commitment Therapy for Mental Health Problems in College Students: A Randomized Controlled Trial | 124 |

Figure 7 offers a visualisation of their bibliographical coupling among documents that had at least 10 citations. Overall, documents were clustered in seven groups.

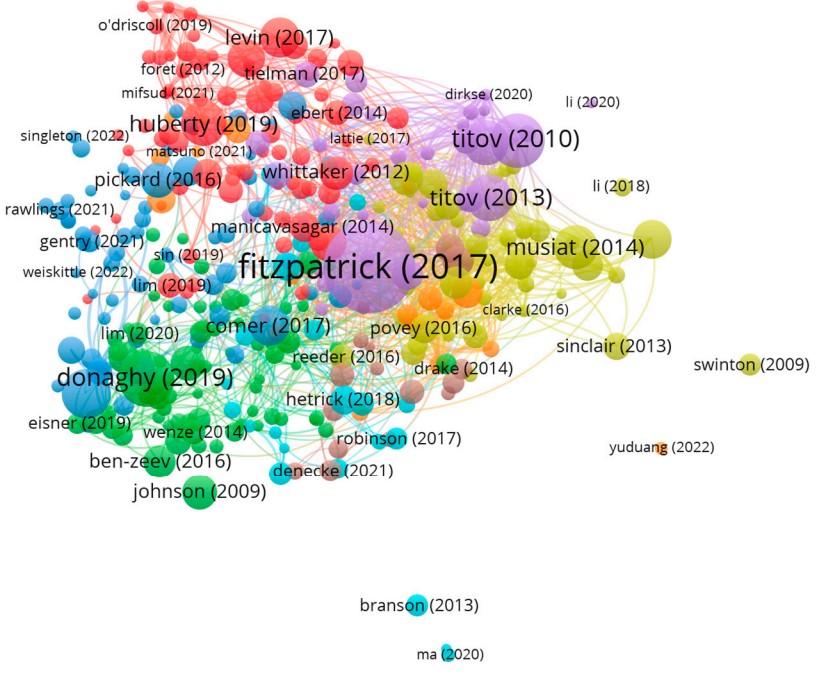

**Figure 7.** Documents' bibliographical coupling.

The largest cluster in red had seventy-seven documents that included studies that reported on measures of acceptability of web-based psychological interventions with mindfulness-based intervention being frequently cited within this cluster. The most cited documents within this cluster was Huberty et al.'s (2019) "*Efficacy of the Mindfulness Meditation Mobile App "Calm" to Reduce Stress Among College Students: Randomized Controlled Trial*" with 163 citations and Levin et al.'s (2017) "*Web-Based Acceptance and Commitment Therapy for Mental Health Problems in College Students: A Randomized Controlled Trial*" with 124 citations.

The second cluster in green had seventy-seven documents and included studies that reported on the acceptability of digital mental health technologies primarily by service users and practitioners in interventions within primary care. The most cited documents within this cluster were Donaghy et al.'s (2019) "*Acceptability, benefits, and challenges of video consulting: a qualitative study in primary care*" with 242 citations, and Johnson et al.'s (2009) "*Computerized ambulatory monitoring in psychiatry: a multi-site collaborative study of acceptability, compliance, and reactivity*" with 94 citations.

The third cluster in blue with sixty-five documents included studies that often focused on the development and delivery of digital mental health-related interventions and reported on intervention acceptability and usability. The most cited documents within this group was Ben-Zeev et al.'s (2013) "*Development and usability testing of FOCUS: A smartphone system for self-management of schizophrenia*" with 189 citations and Comer et al.'s (2017) "*Remotely delivering real-time parent training to the home: An initial randomized trial of Internet-delivered parent–child interaction therapy (I-PCIT)*" with 126 citations.

The fourth cluster in yellow with forty-nine documents and the fifth cluster in purple with thirty-six documents included studies that reported on the acceptability of digital psychological interventions, primarily for the treatment of depression, generalised anxiety, and psychiatric disorders. The yellow cluster included more pilot studies, survey studies, and targeted outcomes that focused explicitly on perceived acceptability. The most cited papers in the yellow cluster were Musiat et al.'s (2014) "*Understanding the acceptability of*

*e-mental health—attitudes and expectations towards computerised self-help treatments for mental health problems*" with 150 citations and Gun et al.'s (2011) "*Acceptability of Internet Treatment of Anxiety and Depression*" with 119 citations. On the other hand, the purple cluster included more randomised controlled trials, where acceptability was reported as a secondary outcome. The four most cited papers in this cluster were among the top 10 most cited papers. Those were Fitzpatrick et al.'s (2017) "*Delivering Cognitive Behavior Therapy to Young Adults With Symptoms of Depression and Anxiety Using a Fully Automated Conversational Agent (Woebot): A Randomized Controlled Trial*" with 655 citations, Titov et al.'s (2010) "*Internet Treatment for Depression: A Randomized Controlled Trial Comparing Clinician* vs. *Technician Assistance*" with 235 citations, Titov et al.'s (2013) "*Improving Adherence and Clinical Outcomes in Self-Guided Internet Treatment for Anxiety and Depression: Randomised Controlled Trial*" with 173 citations, and Robinson et al.'s (2010) "*Internet Treatment for Generalized Anxiety Disorder: A Randomized Controlled Trial Comparing Clinician* vs. *Technician Assistance*" with 162 citations.

The sixth cluster in light blue with twenty-one documents included pilot studies and survey studies that reported on the acceptability and utility of digital mental health tools. The most cited papers with this cluster were Hetrick et al.'s (2018) "*Youth Codesign of a Mobile Phone App to Facilitate Self-Monitoring and Management of Mood Symptoms in Young People With Major Depression, Suicidal Ideation, and Self-Harm*" with 64 citations and Rice et al.'s (2018) "*Moderated online social therapy for depression relapse prevention in young people: pilot study of a 'next generation' online intervention*" with 64 citations.

The seventh cluster in orange with twenty documents included studies that reported on attitudes and perspectives towards e-mental health. The most cited papers within this cluster were Apolinário-Hagen et al.'s (2018) "*Public Attitudes Toward Guided Internet-Based Therapies: Web-Based Survey Study*" with 77 citations and Apolinário-Hagen et al.'s (2017) "*Current Views and Perspectives on E-Mental Health: An Exploratory Survey Study for Understanding Public Attitudes Toward Internet-Based Psychotherapy in Germany*" with 49 citations.

Finally, the eighth cluster in brown with sixteen documents included development, pilot, and usability studies that reported on the acceptability of novel digital mental health tools. The most cited papers within this cluster were Prochaska et al.'s (2021) "*A Therapeutic Relational Agent for Reducing Problematic Substance Use (Woebot): Development and Usability Study*" with 49 citations and Suganuma et al.'s (2018) "*An Embodied Conversational Agent for Unguided Internet-Based Cognitive Behavior Therapy in Preventative Mental Health: Feasibility and Acceptability Pilot Trial*" with 48 citations.

## 5. Discussion

The purpose of this study was to explore the development of research in the acceptability of digital mental health-related interventions. The bibliometric analysis based on the WOS database covered 990 documents published by 5598 authors in 1672 institutions from 2008 to 2023. This is the first study that examined the development of research outputs claiming to address the acceptability of digital mental health-related interventions within their research procedures.

Bibliometric tools can provide insights into the most influential research outputs in a field and highlight emergent areas of focus (Agarwal et al. 2016). The analysis of several metrics such as authors' keywords, article-level metrics, and citation patterns provided details on the key areas of interest and research priorities that have been the key drivers of knowledge production. At the same time, the visual network analyses of publication patterns illustrated the popularity and dominance of different approaches to intervention acceptability.

### 5.1. Publication Trends

Our study findings highlight the increase in publications reporting the acceptability of digital mental health-related interventions and show how technological developments and

research contexts have shaped their evolution. Our study findings confirm the results of previous bibliometric analyses in technology-based treatments in psychology that showed an increase in popularity in the 2010s (Flujas-Contreras et al. 2023). Additionally, this study showed an increase in studies addressing intervention acceptability after 2019, which coincides with the increased focus on the importance of feasibility and pilot studies, also shown by the increase in the publication of guidelines on their conduct and role in intervention development and evaluation (Bowen et al. 2009; Lancaster and Thabane 2019; Skivington et al. 2021). Expectedly, publications focusing on digital mental health interventions increased after the onset of the COVID-19 pandemic, a trend that has also been reported in other bibliometric analyses and reviews of the literature (Ellis et al. 2021; Riboldi et al. 2023). However, there was no further increase in 2023, which may be the outcome of the pandemic pressures in healthcare systems across the world subsiding and as a result, services and research programmes reverted to in-person intervention delivery (Mindsolent n.d.). Intervention acceptability is typically addressed in feasibility studies (Orsmond and Cohn 2015). For this reason, it is expected that the publication of studies reporting on the acceptability of interventions will follow the timeline of technological innovations that allowed the implementation of novel interventions with diverse ways of use engagement. The mapping of technology-related keywords across the studies' average publication years demonstrated how the acceptability of mobile-based mental health interventions has been the dominant area of interest since 2020. The research focus on app-delivered interventions has continued in the later years, while intervention delivery via chatbots has also increased in popularity. A strength of mobile interventions is that they can be personalised to the needs of individual patients and use numerous engagement strategies. However, as shown by recent reviews, there are also numerous user-centered and intervention-specific parameters that influence user satisfaction and engagement with an intervention and the degree to which they are preferable over face-to-face delivery methods (Chan and Honey 2022; Gan et al. 2022). Our study results showed how research activity between 2020 and 2022 was shaped by research conducted in the context of the COVID-19 pandemic. The keyword "COVID-19" was the fifth most frequent keyword within all the documents selected since 2008. It represented unique research content due to the rapid development and implementation of numerous remotely delivered mental health-related interventions either directed toward healthcare professionals or service users (Dominguez-Rodriguez et al. 2022; Witteveen et al. 2022).

*5.2. Visual Network Analyses: Key Findings*

The results of the visual network analyses illustrated the key drivers of research activity that have shaped the research topic. A total of 990 research publications were published in 78 countries. In line with well-documented trends in e-mental health (Helha and Wang 2022; Zale et al. 2021), the main contributors and the authors of the most influential publications were based in high-income countries (namely the USA, Australia, England, Canada, and Germany), with China and India being the most productive among middle-income countries. The co-authorship analysis showed that several clusters of productive research groups are based in Australia, which may reflect specific regional research interests, such as a focus on overcoming barriers to the accessibility or effectiveness of digital mental health services (Balcombe and De Leo 2021). Moreover, the network analysis of co-authorship showed that only a few authors maintained collaborations with authors based in other countries.

A key finding of the visual network analyses conducted was that the most influential research interests and trends in citations were not always aligned with those that drove the highest volume of publications. For example, the results of both the text co-occurrence analysis and the authors' keyword analysis showed that the largest cluster of documents reported studies focusing on intervention development and implementation. Such studies often targeted the needs of specific population groups (e.g., adolescents and veterans) and consequently adopted compatible research designs (e.g., user-centered development and

participatory action research). On the other hand, the results of the studies' bibliographical coupling showed that the most influential studies were those that reported on the efficacy of interventions, while citation patterns were defined by shared interests in specific intervention approaches (e.g., mindfulness-based interventions) and technology acceptance within primary health services. The only exception to this pattern of results appeared to be research addressing the acceptability of digital mental health-related interventions which was conducted in the context of the COVID-19 pandemic. Such observations may reflect differences in research priorities associated with research funding. Previous research has illustrated that the quantity and quality of publications are related to funding allocation (Ebadi and Schiffauerova 2015), while the increased prioritization of funding allocation in studies focusing on technological applications to mental health by large funding bodies has played a role in their growing popularity (Zale et al. 2021). Similarly, the impact of historical events such as the COVID-19 pandemic can lead to rapid adoption of technology (Zale et al. 2021). This is characterised by agility in developing and evaluating interventions, which also requires equally robust intervention development frameworks. However, discussions, even prior to the COVID-19 pandemic, highlight that constraints associated with more traditional approaches to intervention development essentially limit the potential for the real-world impact of digital mental health technologies (Balcombe and De Leo 2021; Torous and Haim 2018)

### 5.3. Acceptability Approaches

Our study results demonstrated that the variation in authors' approaches to acceptability is often associated with the stage of intervention development during which a certain study is conducted. Sekhon et al.'s (2017) define acceptability as "*a multi-faceted construct that reflects to the extent to which people delivering or receiving a healthcare intervention consider it to be appropriate, based on anticipated or experimental cognitive and emotional responses to the intervention*" and distinguish between prospective acceptability (before participation in an intervention), concurrent acceptability (during participation), and retrospective acceptability (after participation). The text analysis of abstracts and titles showed that acceptability approaches could be distinguished into two broad categories: the development or implementation of an intervention where the focus is on user experience and attitudes and the preliminary efficacy, where the focus on intervention outcomes is such that intervention accessibility can be a secondary outcome. The bibliographical coupling demonstrated that four out of the eight clusters had a distinct focus on acceptability, whereas the other half were clustered around intended primary intervention outcomes (e.g., depression) or intervention approaches (e.g., mindfulness interventions). However, the most cited cluster of studies were those that reported acceptability, assessed as part of randomised controlled trials. This is expected as RCT is the most suitable research design for demonstrating the effectiveness of a specific intervention. At the same time, however, within the overall design of a randomised controlled trial, acceptability can be conflated with satisfaction and, thus, pay limited attention to the complexities of acceptability that are more associated with the sustained adoption of an intervention. Recent reviews of the literature, for example, on the implementation and adoption of digital mental health care during the COVID-19 pandemic highlight the importance of addressing real-world parameters that can influence interventions acceptability by different stakeholders (Witteveen et al. 2022).

### 5.4. Limitations

This study has several limitations that need to be acknowledged. First, the article sampling consisted only of articles published on the Web of Science (WOS). Thus, articles published in other sources were not included in the sample. Furthermore, review papers and book chapters were excluded from this study and, consequently, their bibliometric features were not assessed. Although there were no language restrictions in the article selection, most of the documents were written in the English language, which may reflect an innate limitation of selecting documents from the WOS. Moreover, the search for documents

stating "usability" or "acceptability" in their title, abstract, or keywords may mean that some documents that addressed the development of interventions may not have been captured if they did not use those terms. Finally, the use of publication metadata as a point of analysis of publication trends means that analysis of trends can be impacted by potential discrepancies found within the source data (e.g., incorrect characters, misspellings of authors' names, institutions affiliations, omissions of citations, etc.) (Pranckutė 2021).

## 6. Conclusions

Despite these limitations, this study contributes to the analysis of the evolution of the ways that intervention acceptability is understood, assessed, and prioritised in studies of digital mental health-related interventions. Furthermore, the use of the term "digital mental health-related interventions" allowed the inclusion of studies that reported interventions to support general mental well-being and were directed to any type of user. Finally, the comprehensive examination of authors' keywords allowed us to map technology-related terms separately from intervention acceptability terms and, thus, be able to have a clearer view of key drivers of publication interests.

*Implications for Future Research*

The results of this study demonstrated how the study of the acceptability of digital mental health-related interventions has evolved to encompass a large variety of parameters that transcend the different stages of intervention development and evaluation. Future research will need to identify the parameters of acceptability that are addressed in different studies and the degree to which the assessments undertaken adequately address acceptability, and eventually, explore its links with intervention adoption rates and real-world impact. For example, systematic reviews in high-, middle-, and low-income countries can explore the relevant evidence on interventions' acceptability within those contexts and the barriers for their implementation. Moreover, future research will need to explore situations and contexts where digital mental health-related interventions may even extend existing inequalities (Krukowski et al. 2024). Finally, both reviews of existing evidence and empirical research can inform the development of guidelines for intervention acceptability to be adequately reported even in cases where it serves as a secondary outcome to large randomised controlled trials.

**Funding:** This research received no external funding.

**Institutional Review Board Statement:** Ethical review was not required for this study is it did not involve humans.

**Data Availability Statement:** The original contributions presented in the study are included in the article, further inquiries can be directed to the corresponding author.

**Conflicts of Interest:** The author declares no conflicts of interest.

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
