# Peer review of "Research Trends in the Study of Acceptability of Digital Mental Health-Related Interventions: A Bibliometric and Network Visualisation Analysis"

_socsci, doi:10.3390/socsci13020114_

Round 1
Reviewer 1 Report
Comments and Suggestions for Authors
The title and the content of this paper seem to lack congruence. As it is currently titled, the paper suggests it will say something about whether digital based treatments are acceptable or not to practitioners and users. However it does not address this issue. Instead the focus is on publications trends and authors’ approaches. The term bibliometric in the title indicates there will be a statistical analysis of outputs but the title implies this will be done to ascertain what authors/articles say about the actual acceptability of digital interventions.
Otherwise there is not a lot to argue with as the paper is so heavily descriptive. What is the main contribution of the paper? What is new here? A longer discussion/analysis section may have allowed for more reflection on why this matters to future authors (though that issue seems secondary than the actual acceptability of the digital treatments).
Also, some rationale for using only Web of Science would help. Why is this the best database and why did you limit yourself to only this one? (this could be put in the limitations section, or, ideally, sooner).
Small typos: line 305 (should be examined) and line 510 ‘into to’
Author Response
I’d like to thank both reviewers for their constructive feedback. Please see below the detailed responses to their comments:
Reviewer 1:
The title and the content of this paper seem to lack congruence. As it is currently titled, the paper suggests it will say something about whether digital-based treatments are acceptable or not to practitioners and users. However, it does not address this issue. Instead, the focus is on publications' trends and authors’ approaches. The term bibliometric in the title indicates there will be a statistical analysis of outputs but the title implies this will be done to ascertain what authors/articles say about the actual acceptability of digital interventions.
Response: The title has now been revised “Research trends in the Study of Acceptability of Digital Mental Health-Related Interventions: A Bibliometric and Network Visualization Analysis
I acknowledge that it may not have been clear in the paper’s title, but the study’s focus was to explore ways in which the concept of acceptability was operationalized within the context of empirical research looking only at the studies’ bibliometric features.
For this reason, the main objective was revised as “The aim of this study was to explore the knowledge production patterns associated with the concept of acceptability of digital mental health-related interventions in primary research studies”.
Otherwise there is not a lot to argue with as the paper is so heavily descriptive. What is the main contribution of the paper? What is new here? A longer discussion/analysis section may have allowed for more reflection on why this matters to future authors (though that issue seems secondary than the actual acceptability of the digital treatments).
Response: The discussion section has now been updated
Also, some rationale for using only Web of Science would help. Why is this the best database and why did you limit yourself to only this one? (this could be put in the limitations section, or, ideally, sooner).
Response: “The WOSCC was selected as it’s regarded as one of the most authoritative databases, it is more selective than other databases and has greater keyword sensitivity. Although its overall coverage is smaller than that of Scopus, using only one database allows for a better quality of citations analysis”
Reviewer 2 Report
Comments and Suggestions for Authors
Acceptability of Digital Mental Health-Related Interventions: A Bibliometric Analysis
The paper aimed to explore the knowledge on the acceptability of digital mental health-related interventions with bibliometric analysis. The main contribution of the paper is to describe developments in how the acceptability of interventions is understood, assessed, and prioritized in studies of digital mental health interventions. The strength of the paper is the disclosure of trends relevant to the acceptability of interventions.
The paper is relevant to the field of digital mental health research. The importance of research is well justified in the introduction as is the lack of knowledge about the acceptability of digital mental health interventions.
Details given in the methods section were clear. However, there is a need to clarify a few things. First, I suggest specifying the date of the search, because the article is to be reviewed in January 2024 and then the whole of January 2024 could not be included in the review. Second, it would be important to describe on what basis were 10 % of the documents selected for a detailed examination of the concepts of "acceptability" and "usability". Third, is it possible to describe what documents’ bibliographic coupling means and why it was done in the method section? Fourth, could the author briefly justify why only the Web of Science database was used? Last, I recommend the author to check the text, use and meaning of abbreviations, and layout under the heading Data source and search strategy. Also, abbreviations use and meanings should be checked in all the paper.
The tables need unification and clarification. Now the paper seems to use three different table models and, for example, table 3 is difficult to understand. The figures were informative but had to be enlarged on the screen to understand the text. Could you clarify the pictures somehow? Would it be possible to describe all the figures, as the author had done on page 5 Figure 2? This could help in understanding the figures.
​Do the discussion section headings include: publication trends, key findings, approaches to acceptability, limitations, and conclusions? Or do these titles form their separate paragraphs? I think it would be good to clarify this. In the Acceptability Approaches section, the discussion was very good and versatile. The limitations of the study were also well considered. I would also have liked to read the authors' thoughts on the strengths of the study.
From the conclusions, I missed a summary of the implication of the research results in the direction of future research. This was mentioned very briefly at the end of the limitations paragraph. The topic of the study, the acceptability of digital mental health interventions, is important. It would be beneficial if the author would bring out in a little more depth how the research results could be used and what should be paid attention to in the future.
You should go through the text carefully because there were some typos in it, for example on page 8 in line 251 keywordS, on page 11 in line 322 on page 16 in line 485 Covid-10.
Author Response
Details given in the methods section were clear. However, there is a need to clarify a few things. First, I suggest specifying the date of the search, because the article is to be reviewed in January 2024 and then the whole of January 2024 could not be included in the review.
Response: Thank you for bringing this to my attention. January has now been excluded from the review and the relevant graph has been updated.
Second, it would be important to describe on what basis were 10 % of the documents selected for a detailed examination of the concepts of "acceptability" and "usability".
Response: Thank you for the opportunity to discuss this a bit further. That is closely related with two study parameters. The first parameter involves the keyword searches for empirical studies examining the concept of acceptability.
“The keyword searches for papers addressing the concept of acceptability included only the keywords “acceptability” or “usability” as they represent two distinct but greatly interlinked generic concepts. This strategy allowed the inclusion of empirical studies that tap on different ways in which intervention acceptability has been operationalized”.
The second study parameter was that the main focus of the study itself which was essentially to examine what information (if any) can be extracted solely based on bibliometric features about the way acceptability is addressed within the studies. A limitation of any bibliometric analysis is that it reports on documents’ bibliographic metadata rather than effects or outcomes. As intervention acceptability was often reported as a secondary outcome in many of the included studies it was important to decide which bibliometric features would be more suitable to include in the analyses and in what ways. For this reason, it was decided to investigate the descriptions of references to or assessment of acceptability or usability in a sample of the included studies. That was a preliminary step to get an idea of the breadth of the scope of the studies and as a result, inform the bibliometric analysis strategy. For example, a general observation from this preliminary investigation was that differentiations in the way acceptability was operationalized seemed to follow differentiations across different studies’ research designs.
Then, as some titles/abstracts refered to acceptability or usability multiple times and in others only once, it was decided to conduct a co-occurrence analysis of text found in documents’ titles and abstracts based on their binary frequency (their presence or not). If the research focus had been the assessment of either acceptability or usability (e.g. as a feasibility outcome for example) then I believe that a complete data extraction would be advisable.
Third, is it possible to describe what documents’ bibliographic coupling means and why it was done in the method section?
Bibliographic coupling is a similarity measure that asses the frequency in which specific documents are cited together in other publications and as a result it is an indicator of shared research themes
Fourth, could the author briefly justify why only the Web of Science database was used?
“The WOSCC was selected as it’s regarded as one of the most authoritative databases, being more selective than other databases and greater keyword sensitivity. Although its overall coverage is less than Scopus using only one database allows for a better quality of citations analysis”
Last, I recommend the author to check the text, use and meaning of abbreviations, and layout under the heading Data source and search strategy. Also, abbreviations use and meanings should be checked in all the paper.
The tables need unification and clarification. Now the paper seems to use three different table models and, for example, table 3 is difficult to understand. The figures were informative but had to be enlarged on the screen to understand the text. Could you clarify the pictures somehow? Would it be possible to describe all the figures, as the author had done on page 5 Figure 2? This could help in understanding the figures.
Response: The tables have now been formatted to match the journal’s requirements so they should now have similar characteristics and more attention has been given to abbreviations Also, I have tried to enlarge some of the pictures and especially figure 5 so that the names of the keywords can be more easily read.
​Do the discussion section headings include: publication trends, key findings, approaches to acceptability, limitations, and conclusions? Or do these titles form their separate paragraphs? I think it would be good to clarify this. In the Acceptability Approaches section, the discussion was very good and versatile. The limitations of the study were also well considered. I would also have liked to read the authors' thoughts on the strengths of the study.
From the conclusions, I missed a summary of the implication of the research results in the direction of future research. This was mentioned very briefly at the end of the limitations paragraph. The topic of the study, the acceptability of digital mental health interventions, is important. It would be beneficial if the author would bring out in a little more depth how the research results could be used and what should be paid attention to in the future.
Response: The discussion has been further developed including titles on Publication trends, Visual Network Analyses: Key findings, Acceptability approaches, Conclusion, and Implications for future research